# Electrodialysis Metathesis (EDM) Desalination for the Effective Removal of Chloride and Nitrate from Tobacco Extract: The Effect of Membrane Type

**DOI:** 10.3390/membranes13020214

**Published:** 2023-02-09

**Authors:** Qian Chen, Yue Zhou, Shaolin Ge, Ge Liang, Noor Ul Afsar

**Affiliations:** 1Applied Engineering Technology Research Center for Functional Membranes, Institute of Advanced Technology, University of Science and Technology of China, Hefei 230088, China; 2Anhui Provincial Engineering Laboratory of Functional Membrane Materials and Technology, Department of Applied Chemistry, School of Chemistry and Materials Science, University of Science and Technology of China, Hefei 230026, China; 3IAT USTC-AHZY Joint Laboratory of Chemistry & Combustion, Institute of Advanced Technology, University of Science and Technology of China, Hefei 230088, China; 4Anhui Key Laboratory of Tobacco Chemistry, Anhui Tobacco Industrial Co., Ltd., 9 Tianda Road, Hefei 230088, China

**Keywords:** electrodialysis metathesis desalination, tobacco liquor, ion exchange membranes, diffusion dialysis, citric acid

## Abstract

Electrodialysis Metathesis (EDM) desalination was investigated using a squad of three ion-exchange membranes (ACS, TW-A, and A3) and simulated tobacco extract liquid for selective ions removal. We have studied various factors affecting EDM desalination efficiency using a complete experimental design. First, diffusion dialysis (DD) was conducted to determine the permeation rate of different anions in tobacco liquor with different membrane materials. We conclude that A3 had the fastest permeation rate of anions. However, ACS has the lowest permeation rate for different salts. The investigation of the EDM process showed the excellent ion permeation ability of A3 by detecting the current, conductivity, and ion concentration of the target tobacco liquor in the metathesis chamber of the EDM process. The EDM had shown the most excellent chloride ion removal ability. We found that A3 was the best membrane for the EDM process of tobacco liquor.

## 1. Introduction

As one of the major non-food crops, tobacco plays an important role in agriculture; however, tobacco industries produce a large number of by-product wastes, which are not recycled, such as tobacco stems and broken tobacco flakes [1,2,3]. Recycled tobacco is an excellent means of recycling these wastes [4]. Usually, these wastes are recycled by making tobacco extract liquid through the process of concentration, slurry preparation, and then processing the tobacco extract liquid into artificially reconstituted tobacco leaves, which can complete the recycling of tobacco wastes [5,6,7]. Like raw tobacco leaves, many harmful substances are present in tobacco extracts and inorganic ions, such as potassium ions, chloride ions, nitrate, and sulfate [3,8,9]. Studies have shown that chloride and nitrate ions greatly impact cigarettes’ hygroscopic state and burning state [10,11]. At the same time, a large number of inorganic ions will also harm human health [12,13,14]. They can greatly reduce the taste of tobacco inhalation and seriously affect tobacco quality [15,16,17]. For the harmful components in tobacco extracts, various methods have been developed, such as microbial fermentation solvent extraction [18,19], multi-stage extraction [20], freezing and centrifugation, and ultrasonic-assisted extraction.

However, these methods have not achieved satisfactory results in the selective removal of inorganic ions and also suffer from cumbersome steps and additional environmental pollution problems.

In contrast, electrodialysis (ED) has outstanding advantages due to its efficiency and low energy consumption for removing inorganic ions from tobacco extracts [21,22]. Our previous study showed that ED produced better results in the selective removal of inorganic ions from tobacco extracts [23,24]. Bazinet et al. used ED to remove polyphenols from tobacco extracts and explored the effect of different commercial membranes on polyphenol removal, with demineralization rates up to 89.5%. Many researchers have demonstrated that ED with selective ion exchange could deter the potential scaling issue during the concentration procedure [25,26]. Zhang et al. examined the separation efficiency of mono/divalent ions from RO brine employing SED. ED is the best option to enhance water retrieval, nutrient removal, and producing water appropriate for groundwater recharge [25,27,28].

Furthermore, Zhang et al. operated a coupled SED-CED approach to achieve scientific and technological analysis of the salt purity produced from SWRO brine [29]. A novel SED process was designed by Liu et al. to separate/recover the nitrogen and phosphorus nutrients and implement desalination in secondary effluent treatment [30]. However, some problems even exist in the SED. For instance, utilizing brine chemical resources is not facilitative because divalent ions become entangled in the diluting chamber. Therefore, ED-based technologies must be designed to stimulate the separation and pre-concentration of typical brine-containing scaling components. Electrodialysis metathesis (EDM) systems represent a novel reactor based on ED processes, with the remarkable ability to recombine and concentrate ions simultaneously [31]. An EDM stack is structurally composed of two pairs of ion exchange membranes and two pairs of spacers with four plane channels; such systems can also transform sparingly soluble salts into highly soluble liquid salts. By EDM operation, 98.5% of water recovery was reported by some researchers. Further, Cappelle et al. assessed the feasibility of the ZDD process based on EDM as a means of elevated recovery brackish water desalination with a mathematical model. Model outcomes demonstrate that ZDD can achieve above 97% system recovery for brackish water, with a feed TDS concentration of <3 g/L and relatively high fractions of multivalent ions [31].

Considering the efficiency and high productivity, we modified the EDM process with three compartments.

In this work, we report a new electrodialysis–metathesis assembly, which adopts a three-chamber design of desalination–metathesis, and concentration chambers and passes the target tobacco extracts into the metathesis chamber. We used citric acid in the desalination chamber and other substances under an electric field to increase the flavor and taste of tobacco extracts. At the same time, we removed the harmful ions chloride and nitrate in tobacco extracts from the tobacco extracts. We also measured the ionic composition of tobacco extracts and the ion permeation and retention ability of the selected membranes. This work thus delivers a practical direction for the ZDD and reclamation of high-salinity wastewater with a high-scaling prospect, which exists widely in seawater desalination engineering and industrial wastewater treatment processes.

## 2. Experimental Section

### 2.1. Chemical and Materials

The Technology Center of Anhui China Tobacco Industry Crop (Anhui, China) provided the reconstituted tobacco extracts used in the experiments. The chemical reagents, such as potassium chloride, potassium nitrate, potassium phosphate, potassium sulfate, potassium malate, potassium citrate, sodium sulfate, etc., were provided by Sinopharm Chemical Reagent Co., (Shanghai, China) Commercial membranes ACS, TW-A, and A3 anion exchange membranes were kindly supplied by Astom Crop (Tokyo) Japan, Shandong Tianwei Membrane Technology Co., Ltd. (Weifang, China), and Zhongke Xinyang Membrane Technology Co., Ltd., (Anhui, China), respectively.

### 2.2. Experiment

To investigate the permeation ability of different anions across the membranes, we prepared a tobacco solution according to the composition of inorganic ions in the real tobacco extracts. We then measured the concentration of chloride, nitrate, phosphate, sulfate, malate, and citrate ions in the simulation to test the permeation coefficients of membranes. Different concentrations of simulated solution were prepared, i.e., 0.028 mol/L potassium chloride solution, 0.012 mol/L potassium nitrate solution, 0.007 mol/L potassium phosphate solution, 0.016 mol/L potassium sulfate solution, 0.053 mol/L potassium malate solution, and 0.008 mol/L potassium citrate solution. The test was carried out by a static diffusion dialysis test apparatus (Figure 1), where 100 mL of the prepared simulated solution was added to one side of the membrane and 100 mL of DI water to the other. The experiment was run for 1 h under stirring.

The permeation coefficients of different anions were calculated by the following equations
(1)$$U=\frac{M_{i}}{At\Delta C}$$
*i* = represent the anion in question. The diffusion coefficient was measured according to Fick’s law of diffusion:(2)$$J_{i}=-D_{i}\frac{dC_{i}}{dx}$$
(3)$$\frac{dC_{i}}{dx}=\frac{C_{2}-C_{1}}{J}$$
(4)$$D_{i}=\frac{J_{i}.l}{C_{2}-C_{1}}$$

*J_i_* is the ion flux, *D_i_* is the diffusion coefficient of ions in the membrane, *l* is the thickness of the membrane, and *C_2_* and *C*_1_ are the ions concentration on the water and feed sides, respectively. The derivation of the above equations leads to the permeate fluxes of different membranes for different ions with trends corresponding to the permeation coefficients of different membranes.

The membrane stack configuration is: “Anode-[C-AEM1-AEM2]n-C-Cathode”, as shown in Figure 2. This experiment used three pairs of membranes (n = 3). A solution of K3 Cit (potassium citrate, 0.2 M) to the desalination chamber, tobacco extract (solid content about 7%) to the metathesis chamber, 0.1 M KCl solution to the concentration chamber, 0.3 mol/L K_2_SO_4_ solution to the pole chamber, A3 for the anode membrane, AEM1 for the higher-density electrodialysis cathode membrane, and AEM2 for the lower-density electrodialysis cathode membrane. AEM2 was selected as the electrodialysis negative membrane with lower densities. Because the ionic system of tobacco extract is too complex, to further investigate the target ions transport, tobacco simulated solution was prepared for electrodialysis testing under the same experimental conditions, and the tobacco solution ionic system was the same as the diffusion dialysis ionic system.

The experimental process examines the effect of different membranes on the ion passage performance, and the experimental parameters to be examined are shown in Table 1. The experimental process is carried out by the current, the conductivity changes, and the actual changes in ion concentration in each chamber.

## 3. Results and Discussion

The permeation coefficient of different membranes for anions (chloride, nitrate, phosphate, malate, and citrate ions) were tested, and the results are shown in Figure 3. The ion concentration in the diffusion chamber showed a linear relationship and gradually increased with time. In addition, for the same potassium salt solution, the ion concentration in the concentration chamber after static diffusion experiments for four different membranes showed the following trend: A3 > TW-A > ACS. Subsequently, the permeation coefficients of different membranes were calculated, and the results are shown in Figure 3.

For the same potassium salt solution, the order of the magnitude of the permeation coefficient for different electrodialysis was A3 > TW-A > ACS. For the same membrane, the permeation coefficient varied for different salts. For A3, the permeation coefficient of different ions was chloride ions > nitrate ions > sulfate ions > nitrate ions > citrate ions > malate ions, which indicated that the removal of chloride ions was relatively difficult in the tobacco extract solution. For the TW-A membrane, the permeation coefficients of different ions were comparable for nitrate, sulfate, and phosphate ions. Finally, the ACS membrane has the worst permeation for chloride ions and malate ions, with almost no permeation, comparable to permeation for citrate ions and phosphate ions, and relatively high permeation for sulfate ions. Subsequently, the above data were analyzed and extrapolated in detail. This shows that the permeation selectivity of different membrane piles for different ions has some specificity. It can also be seen from the figure that A3 has a higher permeation coefficient. In addition, all four membranes have higher permeation selectivity for chloride ions.

The above equations show that, combined with Fick’s law of diffusion, the diffusion coefficients of different ion exchange membranes for different ions can be calculated based on the permeation coefficients derived from the actual tests. The higher the diffusion coefficient, the more ion exchange capacity of the membrane, and the looser the structure of the membrane, the more solution ions permeate through the membrane. As can be seen in Figure 3, A3 has the fastest permeation of the different ions through this membrane. On the contrary, ACS has the lowest and has poor permeation ability for different salts. The above experimental results can provide good guidance for the subsequent selection of suitable ion exchange membranes to improve the separation effect of electrodialysis.

In examining the effect of membrane type on the performance of exchanged electrodialysis, the A3 membrane was used for the AEM2 anion exchange membrane. In contrast, three different ion exchange membranes were used for the AEM1 electrodialysis anion exchange membrane, namely, the ACS, TW-A, and A3 membranes. Figure 4 shows that the electrodialysis–metathesis stacks with ACS, TW-A, and A3 all have a decrease in the membrane stack current with time during operation. This is because the ion concentration in the desalination chamber decreases as the electrodialysis–metathesis process proceeds.

For the membranes, the comparison between the tobacco simulated (Figure 4a) and tobacco extract (a) shows a gradual decrease in conductivity with time. Still, the pattern of change with time is not the same. The mock tobacco solution changed more slowly in the first 40 min compared to the later period. This is due to the limited conductivity of the membrane and the sufficient number of ions in each chamber to meet the current conductivity, so the first period decreases slowly. According to the change of conductivity of each chamber with time (Figure 4), the fastest conductivity decrease in the tobacco simulated is in the desalination chamber. In the later period, the ion concentration in the desalination chamber becomes very low. The concentration also decreases rapidly, leading to the decreased conductivity of the whole electrodialysis–metathesis device. In the case of the tobacco extract solution, the trend of current change is to decrease rapidly at first, then gradually stabilize in the later stage. It is more complex, and the degree of contamination of the membrane is greater than that of the simulated, resulting in the ion transfer membrane’s ability decreasing continuously with the operation of electrodialysis; this can also be proved by the current situation of the tobacco simulant and tobacco extract solution at 60 min, which is lower than that of the tobacco extract solution at 60 min because there are no more ions in the electrodialysis–metathesis stack fade chamber of the tobacco simulant to conduct the current. The current in the tobacco extract electrodialysis–metathesis stack is lower because the ions cannot pass through the membrane due to the more serious membrane contamination and blockage of ion channels, which leads to a slower reduction of ion concentration in the desalination chamber, so the current in the tobacco extract electrodialysis–metathesis stack will be maintained at a higher level than that in the tobacco simulant electrodialysis–metathesis stack in the later stage.

According to the Nernst–Planck equation, the flux of ions in the electrodialysis–metathesis process is proportional to the potential difference or current density on both sides of the membrane. Therefore, when the membrane used makes the electrodialysis–metathesis stack have a higher current density, the salt flux of the electrodialysis–metathesis will be higher, the conductivity of the desalination chamber will decrease faster, and the conductivity of the concentration chamber will increase faster, as shown in Figure 5.
(5)$$J_{i}=-D_{i}\frac{dC_{i}}{dx}-D_{i}\frac{z_{i}C_{i}F}{RT}\;\frac{\partial \varphi }{\partial x}$$

In addition, it can be seen from Figure 5 that the conductivity of the stock solution in the metathesis chamber decreases with the treatment of the mock tobacco solution by the electrodialysis–metathesis process because the organic acid anions migrating from the desalination chamber to the metathesis chamber replace the harmful inorganic anions (Cl^−^ and NO_3_^−^) in the stock solution during the electrodialysis–metathesis process, which leads to a decrease in the conductivity in the metathesis chamber.

To obtain more detailed information, we measured the ion concentrations of tobacco extracts during the electrodialysis–metathesis process, as given in Figure 6a–c. Since the purpose of the experiment was to examine the electrodialysis–metathesis device for the removal of harmful ions from tobacco liquor, potassium retention, and the introduction of organic acid roots, we measured the potassium ion concentration, chloride ion concentration, and potassium citrate concentration of tobacco liquor in the metathesis chamber.

The potassium ion concentration in each chamber of the electrodialysis–metathesis stack under different membranes shows that among the ionic membranes, the A3 membrane has the best retention ability for potassium ions, and the potassium ion concentration in the metathesis chamber always remains above 0.1 M. In contrast, both the ACS and TW-A membranes have poor retention ability for potassium ions. Overall, the potassium concentration in the desalination chamber showed a decreasing trend. The potassium concentration in the concentration chamber showed an increasing trend because the potassium in the desalination chamber assumed the role of carrying current under the electric field and entered the electrode chamber through the anode membrane. In contrast, the potassium in the concentration chamber originated from the electrode chamber. Among the membranes, the potassium ion retention ability ranking when using different membranes was: A3 > TW-A > ACS.

The citrate ion concentration in each chamber of the electrodialysis–metathesis stack under different AEM2 membranes is shown in Figure 7. During the operation of electrodialysis–metathesis, the citrate ion concentration in the concentration chamber shows a trend of decreasing to different degrees, which is because the negatively charged citrate ion will continuously migrate through AEM2 and AEM1 membranes to the anode chamber under the action of the electric field. We hope that the citrate heel ion will stay in the metathesis chamber to increase the flavor of tobacco liquor when the membrane continuously passes through AEM2 membrane into the metathesis chamber. Figure 7 shows that when the AEM1 membrane is A3 and ACS, the metathesis chamber is enriched with more citrate ions, reaching 0.0699 M and 0.0683 M, respectively. When other membrane materials were used, the concentration of citrate heel even showed a decreasing trend, which indicates that the other membranes have a more inferior ability to block citrate, which also means that these membranes do not play a role in the flavor enhancement of tobacco liquor during practical application.

The citrate ion concentration in each chamber of the electrodialysis–metathesis stack under different AEM2 membranes is shown in Figure 8. Among the different AEM2 membranes, the chloride ion in the metathesis chamber has a better removal effect, which is because among all the ions, chloride ion exists with less water and energy and a smaller ion radius, which is the easiest to pass through the membrane, so its passage under the action of current is also the best, firstly, as the ion carrying the current. The ion concentration in the metathesis chamber shows a significant reduction trend. The ion concentration in the desalination chamber tends to be 0 due to the absence of chloride ions in the initial solution. In contrast, the ion concentration in the concentration chamber shows a rising trend with the electrodialysis operation. Finally, the removal effect of chloride ions was ranked from good to bad: A3 > TW-A > ACS.

The ion chromatography results yielded the effects of different ion exchange membranes on removing chloride and nitrate, retaining potassium ions, and introducing citrate, as shown in Table 2. It can be seen that the different membranes achieved more than 80% removal of chloride and nitrate, and in particular, the A3 membrane achieved more than 95% removal of both ions. Moreover, A3 had a relatively good potassium ion retention of 77.12%, while the other two membranes had less than 50% retention of potassium ions. In the investigation of the introduction of organic acid ions, it was found that ACS did not affect the introduction of citrate ions. Still, even the original citrate ions present in the tobacco liquor were removed. At the same time, the A3 and TW-A membranes, on the contrary, both showed excellent increased citrate ions, up to 249.75% and 241.50%, respectively. Such a large amount of citrate ion introduction can greatly improve the taste and flavor of tobacco liquor.

In a comprehensive evaluation, A3 membranes were used as AEM2 membranes in the electrodialysis–metathesis operation to obtain excellent results of harmful ion removal, beneficial ion retention, and organic acid ion introduction at the same time.

## 4. Conclusions

We investigated the effect of different membranes on the removal of chloride, nitrate ions, and potassium retention of tobacco liquor, and the addition of organic acid ions in the electrodialysis–metathesis process. First, diffusion dialysis studies were conducted on three selected membranes to investigate the permeability of common ions in tobacco liquor. The A3 membrane had the highest ion permeation coefficient for different ions. On the contrary, ACS has the lowest ion permeation coefficient and has a poor permeation capacity for ions. The actual investigation of the electrodialysis–metathesis process was also conducted, and the ion permeation ability of A3 was found by detecting the current and the conductivity data between different chambers. The ion concentration of the target tobacco liquor in the metathesis chamber was monitored, and it was observed that the EDM process had the highest efficiency for chloride removal, citrate introduction, and the ion permeation of the original tobacco liquor for A3 membrane. It was noted that A3 was the best membrane for the EDM process of tobacco liquor.

## Figures and Tables

**Figure 1 membranes-13-00214-f001:**
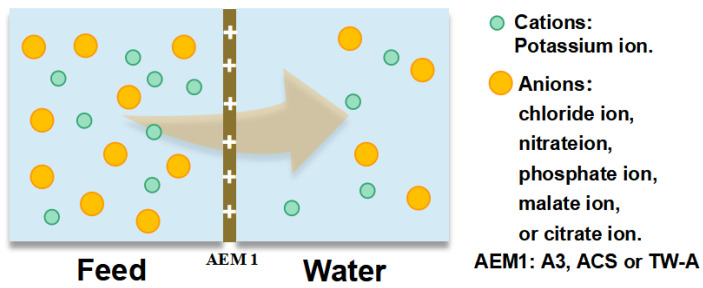
Static diffusion dialysis device to test the permeation coefficients of different anions.

**Figure 2 membranes-13-00214-f002:**
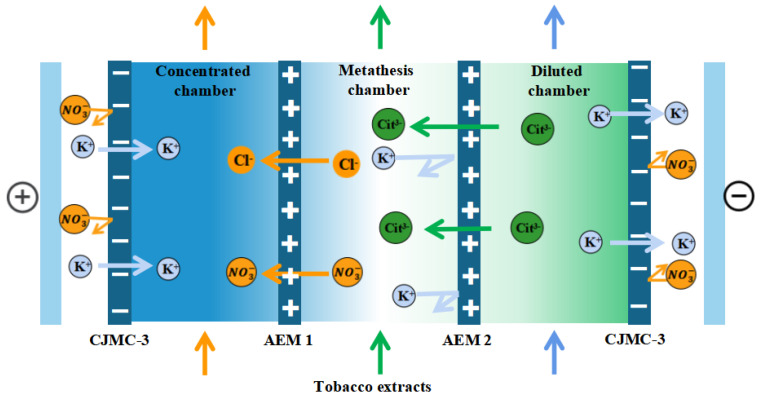
Schematic diagram of electrodialysis–metathesis process for selective remove of anions from tobacco extracts liquid.

**Figure 3 membranes-13-00214-f003:**
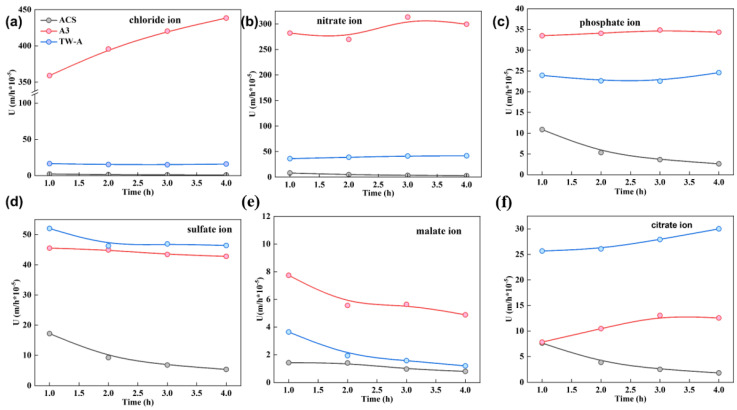
Permeation coefficients of (**a**) Chloride ion; (**b**) nitrate ion; (**c**) phosphate ion; (**d**) sulfate ion; (**e**) malate ion, and (**f**) citrate ion versus time during static diffusion dialysis experiment.

**Figure 4 membranes-13-00214-f004:**
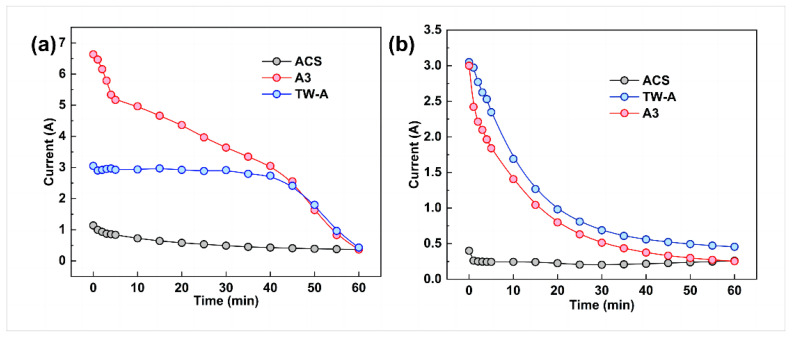
Running current versus time for electrodialysis–metathesis stack tobacco simulant (**a**) under different membranes and tobacco extracts (**b**).

**Figure 5 membranes-13-00214-f005:**
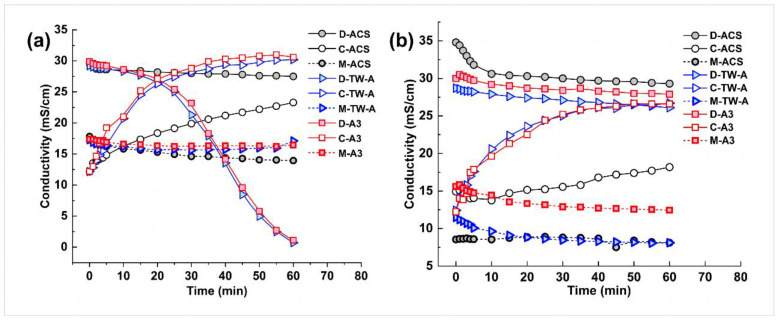
Conductivity versus time for desalination chamber (D), concentration chamber (C), and metathesis chamber (M) for different anion membranes (**a**) for tobacco simulants Tobacco extract; conductivity versus time for desalination chamber (D), concentration chamber (C), and metathesis chamber (M) for different anion membranes (**b**).

**Figure 6 membranes-13-00214-f006:**
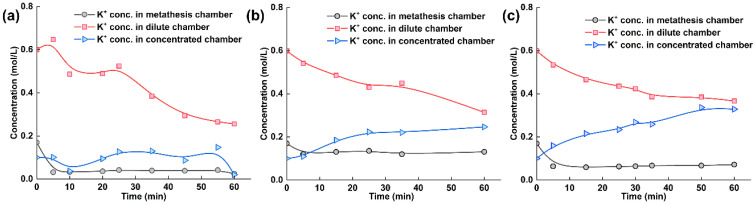
(**a**) Experimental results with ACS membrane and tobacco liquor; (**b**) Experimental results with A3 membrane and tobacco liquor; (**c**) Experimental results with TW-A membrane and tobacco liquor.

**Figure 7 membranes-13-00214-f007:**
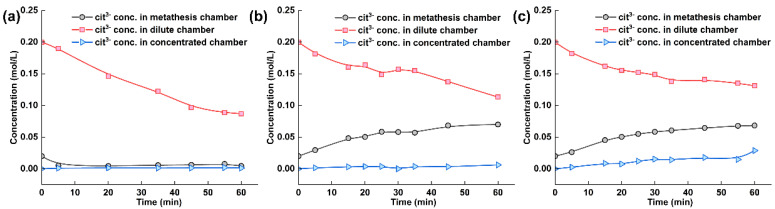
(**a**) Experimental results of tobacco liquor with ACS membrane, (**b**) with A3 membrane, and (**c**) TW-A membrane.

**Figure 8 membranes-13-00214-f008:**
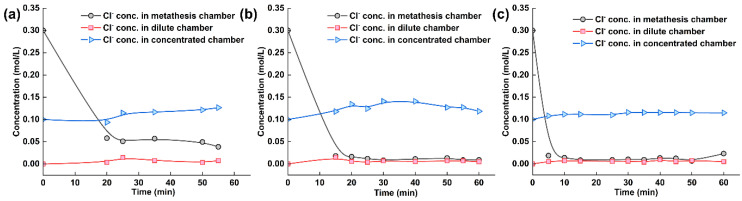
(**a**) Experimental results of tobacco liquor with ACS membrane, (**b**) with A-3 membrane, and (**c**) with TW-A membrane.

**Table 1 membranes-13-00214-t001:** Experimental parameters to be investigated for the electrodialysis–metathesis process.

Parameters	Variables		Other Conditions
**Membrane Type**	AEM1	AEM2	Diluted chamber: 0.2 mol/L K_3_ Cit. Voltage: 7 V Flow velocity: 3 cm/s
	ACS	A3	
	TW-A	A3	
	A3	A3	

**Table 2 membranes-13-00214-t002:** Removal, retention, and introduction of target ions by different anion exchange membranes.

	ACS	A3	TW-A
Cl^−^ Removal rate (%)	87.15	96.97	92.28
NO_3_^−^ Removal rate (%)	90.39	96.11	99.28
K^+^ Retention rate (%)	14.51	77.12	42.17
Cit^3−^ Introduction rate (%)	-78.00	249.75	241.50

## Data Availability

Not applicable.

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
