# Peer review of "Electrodialysis Metathesis (EDM) Desalination for the Effective Removal of Chloride and Nitrate from Tobacco Extract: The Effect of Membrane Type"

_membranes, 2023, doi:10.3390/membranes13020214_

Round 1

Reviewer 1 Report

1. why is affiliation signed as a,b,c,d,...and institutions signed 1,2,3,4...?

2. it is written in the abstract (line 17), that there are repeating squads of 3 ion exchange membranes - it is missleading, as far as i understand there were 3 experiments with 3 different ion exchange membranes, so pls explain it more clearly (it seems that there was 1 squad with 3 different membranes)

3. line 300 - "the better" - correct english

4. figure 1 - pls insert better describing picture of dialysis, this one is unaccaptable

Pls revise the paper according to this notes, there could be many more notes like this, so go through the whole paper one more time, sentence after sentence.

Author Response

  1. 1. why is affiliation signed as a, b, c, d, and institutions signed 1, 2, 3, 4...?

Ans. We are sorry for this unintentional mistake. We have corrected and assigned only alphabets. The author’s list is updated.  

  1. 2. It is written in the abstract (line 17), that there are repeating squads of 3 ion exchange membranes - it is misleading, as far as I understand there were 3 experiments with 3 different ion exchange membranes, so pls explain it more clearly (it seems that there was 1 squad with 3 different membranes).

Ans. Yes, you are right. There is one squad of membranes comprising three different types of membranes i.e., three ion-exchange membranes (ACS, TW-A, and A3).

  1. 3. line 300 - "the better" - correct English.

Ans. Thank you for highlighting this mistake. We have revised the manuscript and corrected all the grammar mistakes to the best of our knowledge. Please see the changes made in red in the revised manuscript.

  1. 4. figure 1 - pls insert better describing picture of dialysis, this one is unacceptable.

Ans. Thank you for your advice. We have revised all Figs and added in the revised manuscript. Please find the description and working mechanism of the DD process.

  1. Pls revise the paper according to this note, there could be many more notes like this, so go through the whole paper one more time, sentence after sentence.

Ans. Thank you very much for your expert advice. We have revised the whole article and tried our best to make it productive and comprehensive for readers. Please see the abstract, introduction and conclusion.

Reviewer 2 Report

Some remarks are in the attached file.

Author Response

The topic is interesting for the application of electrodialysis in tobacco industry and refers to the profile of the journal. However, this manuscript is not suitable for publication. I would recommend to reject it. And bad English is not the main reason. As per my mind, this manuscript does not contribute something new to the theory and practice of electrodialysis. It is only weak attempt of comparison of three membranes permeability, without consideration the influence of these membranes structure and properties. The analysis of membranes special features is required for such study. Besides, one can find many discrepancies and mistakes in the manuscript.

Response. Thank you very much for your remarks. We respect your decision on this manuscript; however, we tried our level best to answer the following questions;

  1. 1. There are no clear schemes of the cells for dialysis and electrodialysis; figures and signatures for them are not informative. The parameters of the cells, the equipment used are not discussed in the text. I cannot see “dialysis device” in Figure 1.

Ans. Thank you for your remarks. We have added clear illustration diagrams for DD and ED and explained the working principle of these processes separately. Please see Fig. 1 and Fig. 2.

Q 2. There is no list of symbols, some symbols are not described in the text. For example, symbols Mi, A, t in equation (1) for U. However, the authors indicated dimension of U as m/h in Fig.3. Lines 154-158. The authors wrote: “As can be seen in Fig.3, A3 has the highest ion diffusion coefficient…”. At the same time, figure 3 has the signature “Dialysis coefficients of various anions…” but the dependence in this figure is for U value (according to equation (1)).

Ans. We agree with your observation. We have added a separate list of abbreviations, symbols, and units; that is being used in this article. Please see page -1. We have corrected and the U here represents the permeation coefficient. Please find the replaced values in the corresponding Fig. 3.

  1. 3. The confusion. It is necessary to mention that in the case of semipermeable membrane one can measure the permeability coefficient but not diffusion coefficient because C1 and C2 – are concentrations in the solutions. (Equations 2, 4 in the manuscript include D in the membrane. Line 96.).

Ans. We respect your critiques; we revised the manuscript and replaced the diffusion constant with the permeation constant. First, we measure the diffusion coefficients of membranes for different ions. Later, we calculated the permeation constant of different ions across the membranes.

  1. 4. The authors use incorrect terms. Example is “Introduction rate” having the value 249% (Table 2).?? Unfortunately, it is possible to continue this list.

Ans. Thank you for your comments. We are sorry for some critical mistakes. We have revised the manuscript.

  1. 5. The article (including Abstract and Conclusions) is not meaningful. The Introduction should preferably analyse works in the field of electrodialysis metathesis.

Ans. Thank you for your comments. We have revised the abstract and conclusion to make them meaningful. Please see both sections in the revised manuscript.    

Reviewer 3 Report

This study details the effect of different membranes on the remove of chloride and nitrate ions and potassium retention of tobacco liquor and the introduction of organic acid ions in the electrodialysis-metathesis process. Authors conducted all the experiments in the scientific way and the results are finely discussed considering reported literature. This manuscript can be considered for publication after some minor revisions, such as:

Please signify the significance of this study in last paragraph of introduction.

Some discussion on existing treatment technologies can be added in the introduction. Following articles can help in improving introduction:

Shahid et al. (2021). Current advances in treatment technologies for removal of emerging contaminants from water–A critical review. Coordination Chemistry Reviews, 442, 213993.

Rout et al. (2021). Nutrient removal from domestic wastewater: A comprehensive review on conventional and advanced technologies. Journal of Environmental Management, 296, 113246.

Please enhance the resolution of figures used in this draft.

Author Response

This study details the effect of different membranes on the remove of chloride and nitrate ions and potassium retention of tobacco liquor and the introduction of organic acid ions in the electrodialysis-metathesis process. Authors conducted all the experiments in the scientific way and the results are finely discussed considering reported literature. This manuscript can be considered for publication after some minor revisions, such as:

Response. Thank you very much for your encouraging remarks. Our response to each query is as follows.

  1. 1. Please signify the significance of this study in last paragraph of introduction.

Ans. Thank you very much for your suggestion. We have added a conclusive para to highlight the significance of this study. Please see the revised manuscript.

  1. 2. Some discussion on existing treatment technologies can be added in the introduction. Following articles can help in improving introduction:

Ans. Thank you very much for your suggestions. The recommended articles are noteworthy and we have added to our article.  

  1. 3. Shahid et al. (2021). Current advances in treatment technologies for removal of emerging contaminants from water–A critical review. Coordination Chemistry Reviews, 442, 213993.
  2. 4. Rout et al. (2021). Nutrient removal from domestic wastewater: A comprehensive review on conventional and advanced technologies. Journal of Environmental Management, 296, 113246.

Ans. Thank you for suggestions. We added the suggested articles in the revised manuscript. Please see ref. 27 and 28.

  1. 5. Please enhance the resolution of figures used in this draft.

Ans. We appreciate your suggestions. We have revised all Figs in this manuscript and improved its resolution.     

Round 2

Reviewer 2 Report

The article has been revised by the authors. It can be published after editing of the text and the improvement of English.

Examples:

Line 9 - Bad English

Lines 10, 23 - very Bad English ("dilute room", "metathesis room"....)

Figure 7 and Table 2. Check the captures - Bad English.

Membrane can not remove or introduce ions... It performs in the system under the effect of the potential gradient. Please, rephrase.

........... 

Line 6. Misprint

Please, check the text again sentence by sentence.